# When do apples stop growing, and why does it matter?

**Maria D. Christodoulou** [1,2], **Alastair Culham**[2]*

**1** Department of Statistics, University of Oxford, Oxford, United Kingdom, **2** University of Reading Herbarium, School of Biological Sciences, University of Reading, Whiteknights, Reading, United Kingdom

* a.culham@reading.ac.uk

**Data Availability Statement:** All relevant data are within the manuscript.

**Funding:** This work was supported by the Biotechnology and Biological Sciences Research Council (BBSRC) as a PhD CASE Studentship 1132848 (MDC), https://www.bbsrc.ac.uk/. The

## Abstract

Apples in the commercial food chain are harvested up to two weeks before maturity. We explore apple fruit development through the growing season to establish the point at which physical features differentiating those cultivars become evident. This is relevant both for the understanding of the growing process and to ensure that any identification and classification tools can be used both on ripened-on-tree and stored fruit. Current literature presents some contradictory findings on apple growth, we studied 12 apple cultivars in the Brogdale National Fruit Collection, UK over two seasons to establish patterns of growth. Fruit were sampled at regular time points throughout the growing season and four morphometrics (maximum length, maximum diameter, weight, and centroid size) were collected. These were regressed against growing degree days in order to appropriately describe the growth pattern observed. All four morphometrics were adequately described using log-log linear regressions, with adjusted $R^2$ estimates ranging from 78.3% (maximum length) to 86.7% (weight). For all four morphometrics, a 10% increase in growing degree days was associated with a 1% increase in the morphometric. Our findings refine previous work presenting rapid early growth followed by a plateau in later stages of development and contrast with published expo-linear models. We established that apples harvested for commercial storage purposes, two weeks prior to maturity, showed only a modest decrease in size compared with ripened-on-tree fruit, demonstrating that size morphometric approaches are appropriate for classification of apple fruit at point of harvest.

## Introduction

Describing fruit growth for apples (*Malus domestica* (Suckow) Borkh.) is both interesting in its own merits, relevant for modern classification methods, and important for commercial fruit trade. Christodoulou et al. [1] demonstrated that mature apples can be correctly classified to cultivar with a 78% accuracy using only external morphometric characters such as length, weight, or color. In countries such as the UK, the constant high demand for apples throughout the year leads to suppliers either importing fruit or harvesting it early and storing it under industrial conditions such as in controlled atmosphere or in hypobaric rooms [2,3]. Standard practice in the industry for apples intended for storage is to harvest before ripening and ripen

funders had no role in study design, data collection and analysis, decision to publish, or preparation of the manuscript.

**Competing interests:** The authors have declared that no competing interests exist.

in storage, as there are no harvest maturity industry standards [4]. We note here that "ripeness" and "maturity" are inconsistently defined terms in the literature [5,6] we use them interchangeably. How early the fruit needs to be harvested depends on cultivar, and expected duration in storage conditions. One aspect that was not addressed by Christodoulou et al. [1] was the impact of the exclusive use of ripened-on-tree (mature) fruit in the analysis. If a classification tool for authenticity purposes is to perform under commercial conditions, it needs to be able to accommodate both stored and tree-ripened fruit. It is therefore crucial to obtain an understanding of the growth pattern for apples throughout the season, in order to be able to explore the difference in basic morphometrics between fruit that has been harvested at different stages of maturation.

Fruit growth recorded as a single metric, such as length or diameter, has been extensively investigated since the beginning of the previous century. According to Bollard [7] the task is more challenging for apples than stone fruit, as different growth patterns have been recorded on the same fruit depending on the growth axis studied, with longitudinal growth slower than transverse. In reality, fruit shape can continue to change until maturity [8]. As a consequence, measurements such as volume and weight may be more appropriate to the overall study of fruit growth.

Most of the research on apple fruit growth is based on studies of single cultivars and suggests that a sigmoid shaped growth curve may suit the description of size characters such as length, diameter or weight. This pattern has been described, amongst others, by Denne [9,10], Orlandini and Moriondo [11] and more recently by Atay et al. [12] and Stajnko et al. [13]. This growth pattern is similar to those described on other pip fruits such as pears [7]. The early growth stage, where cell division is prominent, corresponds to the slow increase of the sigmoid curve at the beginning; the cell expansion to the exponential increase of the curve, and the final plateau to the end of cell expansion and the beginning of the maturation process.

This view has been challenged in the literature at least four times. Firstly, when presenting fruit volume and diameter measurements in terms of days from anthesis, Tukey and Young [14] described a linear growth pattern. The accompanying plots however do not support this view, presenting a sigmoid growth pattern. They interpreted this by suggesting the final plateau observed in their plots was due to weather conditions for the year studied. The second time was by Blanpied [15], who agreed with the curvilinear description by Tukey and Young [14] but presented unclear evidence as the data were all averaged without any indication of the variation present at each time point.

The third time this view was challenged was in the work of Lakso et al. [16] who suggested that a combination of linear and exponential curves fits the weight growth pattern best. To fit their data, they proposed a model originally described by Goudriaan and Monteith [17]. In the original model, factors such as leaf area were potential limitations to the growth curve. Applying this to apple growth, Lakso et al. [16] defined a new model with an early curvilinear stage and late linear stage. Although the model appeared to fit their weight data smoothly, the error bars presented in their plots increased steadily, suggesting possible non-normality of errors. As the paper does not clearly state heteroscedasticity testing prior to modelling, it can be argued that the weight data required transformation prior to analysis.

Finally, Saei et al. [18] proposed a linear bi-exponential model, which can be described as a more generalised version of the Goudriaan and Monteith [17] model, as best fitting diameter and volume data for apples. By directly applying the model described by Buchwald and Sveiczer [19,20] for yeast growth, they suggested that apple growth was best described as a combination of two linear models. As no error bars were presented in the accompanying figures it is difficult to establish if initial modelling assumptions such as error normality were satisfied.

**Table 1. Summary of growth studies for apple development.**

| Author | Morphometric | Growth curve |
|---|---|---|
| Tukey and Young [14] | Diameter and volume | Linear |
| Denne [9] | Diameter and weight | Sigmoid |
| Blanpied (1966) | Weight, volume | Curvilinear |
| Lakso et al. [16] | Weight | Expolinear |
| Orlandini and Moriondo [11] | Diameter | Sigmoid |
| Saei et al. [18] | Diameter, length, and volume | Linear bi-exponential |
| Atay et al. [12] | Diameter | Sigmoid |
| Stajnko et al. [13] | Diameter | Sigmoid |

Morphometric studied as well as proposed growth curve type are presented for each author.

The proposed models and morphometrics presented by the above authors are summarized in Table 1.

A sigmoid curve is probably one of the most widely used models for growth traditionally, both in agriculture and in other fields of biology—a common example being the result of a logistic function. It follows a characteristic S-shape, starting with a slow rate of change, followed by a rapid one, and ending in a slow rate of change. Curvilinear and expolinear curves are very similar, the first one being a more generalized shape of the second one. The growth under both of these models begins as a curve—specifically an exponential curve in the second case—and is followed by a linear segment. Finally, a linear bi-exponential is the combination of two exponential parts and a linear one. A point of particular interest here ought to be the trade-off between the number of parameters required to adequately describe the shape, and the overall fit of the model. This can be evaluated using any of the available information criteria, which take into consideration the number of variables versus the overall fit.

In this work we explore the growth process for four morphometrics on 12 apple cultivars during two growing seasons. We use our findings to compare against the occasionally contradictory published growth curves for apples for insight and to explore how early harvesting for storage purposes is reflected in the morphometric characters of study.

## Materials & methods

### Sample collection

Samples were collected during the 2013 and 2014 growing seasons, from the National Fruit Collection, in Brogdale, Kent, UK. All cultivars were grafted on M9 rootstock. To include a range of shapes and sizes, 12 cultivars belonging to different categories of the IBPG (1982) Apple descriptor guide were chosen (Table 2) for the 2013 harvest. The selection included representatives from each shape grouping from the main harvest season as well as some early-season cultivars. In 2014, replicate sampling of six of the original 12 cultivars was conducted as verification. These were selected using an R randomly generated sequence [21]. A summary of the cultivars sampled is presented in Table 2.

The time of anthesis—defined as a minimum of 50% of flowers open—was visually estimated for the whole orchard for each year. As there was conflicting literature on growth and early fruit growth, increased sampling was carried out during that time. To avoid over-thinning trees through experimental harvest, eight harvesting times were chosen, with ten fruit (five per tree) sampled from each cultivar at each harvesting time with the exception of 20 fruit sampled during final harvest. Each harvested fruit was selected using three randomly

**Table 2. Shape descriptions of selected cultivars.**

| Cultivar | IBPG Shape Grouping | NFC Crowning Type | Year Sampled | Season |
|---|---|---|---|---|
| Adam's Pearmain | Truncate/Conical | Moderate | 2013 | Main |
| Beacon | Conical | Low | 2013 | Early |
| Boiken | Flat | High | 2013, 2014 | Main |
| Bovarde | Oblong | Moderate | 2013, 2014 | Main |
| Catshead | Globose/Conical | Moderate | 2013 | Main |
| Fuji | Ellipsoid | Moderate | 2013, 2014 | Main |
| Kaiser Franz Joseph | Broad/Globose/Conical | High | 2013, 2014 | Main |
| Limoncella | Conical | Low | 2013 | Main |
| Present van Engeland | Conical | Low | 2013, 2014 | Main |
| Red Fortune | Globose | Low | 2013 | Early |
| Rheinischer Krummstiel | Ellipsoid | Low | 2013 | Main |
| Wheelers Russet | Globose | Low | 2013, 2014 | Main |

Twelve cultivars were selected based on their shape description, season duration, and sampling time. Crowning type categories are as defined in the National Fruit Collection Database.

generated sequences, establishing the position on the tree, the branch height, and fruit number on the branch respectively. All random sequences were generated using R [21]. Sampling dates per cultivar are summarized in Table 3. To our knowledge this is the largest number of cultivars studied for apple size development, with a total number 1415 samples.

## Weather data

Weather data were obtained from the MIDAS dataset available through the Met Office [22]. The nearest weather station to the National Fruit Collection was Faversham (Latitude: 51.2965, Longitude: 00.8796, Source ID: 757). Maximum air temperature was used to calculate growing degree days from anthesis using base temperature of 5 ˚C (GDD). Base temperature of 10 ˚C (GDD) was also explored. Growing degree days were calculated using the "pollen" package in RStudio [23,24]. All code used is available in the github repository linked as Supplemental Information.

## Morphometric data collection

Measurements of maximum length and maximum diameter were collected using Vernier callipers; weight was measured using precision scales; and centroid size was collected using six landmarks on the outline of the side view of the fruit as described in Christodoulou et al. [1]. Linear morphometrics were collected for both seasons, geometric morphometrics were only collected for 2013.

## Size development analysis

As part of our preliminary analysis, and to assess whether biennial yield differences may impact findings, a comparison of weights for each cultivar for the final harvest week was conducted using a Welch's two sample t-test and no significant differences were found. Each morphometric was plotted against growing degree days and heteroskedasticity was observed throughout. This was resolved by using two modelling approaches—log-log linear regression, and asymptotic regression on the logarithmically transformed morphometric. The models

**Table 3. Sampling dates per cultivar.**

| | Days from anthesis | Adam's Pearmain | Beacon | Boiken | Bovarde | Catshead | Fuji | Kaiser Franz Joseph | Limoncella | Present van Engeland | Red Fortune | Rheinscher Krummstiel | Wheeler's Russet |
|---|---|---|---|---|---|---|---|---|---|---|---|---|---|
| 10/06/ 2013 | 0 | Estimated Orchard Anthesis | | | | | | | | | | | |
| 24/06/ 2013 | 14 | ✓ | ✓ | ✓ | ✓ | ✓ | ✓ | ✓ | ✓ | ✓ | ✓ | ✓ | ✓ |
| 01/07/ 2013 | 21 | ✓ | ✓ | ✓ | ✓ | ✓ | ✓ | ✓ | ✓ | ✓ | ✓ | ✓ | ✓ |
| 08/07/ 2013 | 28 | ✓ | ✓ | ✓ | ✓ | ✓ | ✓ | ✓ | ✓ | ✓ | ✓ | ✓ | ✓ |
| 15/07/ 2013 | 35 | ✓ | ✓ | ✓ | ✓ | ✓ | ✓ | ✓ | ✓ | ✓ | ✓ | ✓ | ✓ |
| 29/07/ 2013 | 49 | ✓ | ✓ | ✓ | ✓ | ✓ | ✓ | ✓ | ✓ | ✓ | ✓ | ✓ | ✓ |
| 12/08/ 2013 | 63 | ✓ | ✓ | ✓ | ✓ | ✓ | ✓ | ✓ | ✓ | ✓ | ✓ | ✓ | ✓ |
| 02/09/ 2013 | 84 | ✓ | ✓ | ✓ | ✓ | ✓ | ✓ | ✓ | ✓ | ✓ | ✓ | ✓ | ✓ |
| 23/09/ 2013* | 105 | ✓ | | | | ✓ | | ✓ | | ✓ | | | |
| 07/10/ 2013* | 119 | | | ✓ | ✓ | | ✓ | | ✓ | | | ✓ | ✓ |
| 15/05/ 2014 | 0 | Estimated Orchard Anthesis | | | | | | | | | | | |
| 05/06/ 2014 | 21 | | | ✓ | ✓ | | ✓ | ✓ | | ✓ | | | ✓ |
| 25/06/ 2014 | 41 | | | ✓ | ✓ | | ✓ | ✓ | | ✓ | | | ✓ |
| 17/07/ 2014 | 63 | | | ✓ | ✓ | | ✓ | ✓ | | ✓ | | | ✓ |
| 12/08/ 2014 | 89 | | | ✓ | ✓ | | ✓ | ✓ | | ✓ | | | ✓ |
| 22/09/ 2014* | 130 | | | ✓ | ✓ | | ✓ | ✓ | | ✓ | | | ✓ |

Using estimated orchard anthesis time, samples were collected from each selected cultivar at regular intervals. Ten fruit per cultivar were sampled for each timing with the exception of 20 fruit for the last harvest (date indicated by *) of the season for each. Six cultivars were repeated in 2014.

were compared using the Bayesian Information Criterion (BIC), selecting the one with the lowest value [25].

## Results

Log-log linear and asymptotic regression fits for each of the four morphometrics are illustrated in Figs 1 and 2. As base 10 ˚C (GDD) presented the same results to base 5 ˚C we have shown the results for GDD5 and have left GDD10 in the supplementary scripts.

BIC values for each model are summarized in Table 4.

As suggested by BIC, all four of the morphometrics are better represented by log-log linear regressions. The log-log linear regression presented here has the advantages of being easily interpretable, using a low number of variables, and easy to fit. Its disadvantages reflect the primary assumptions of linear regression overall which are normality, linearity, homoskedasticity, and independence. In our particular case, had the log-log transformation not corrected

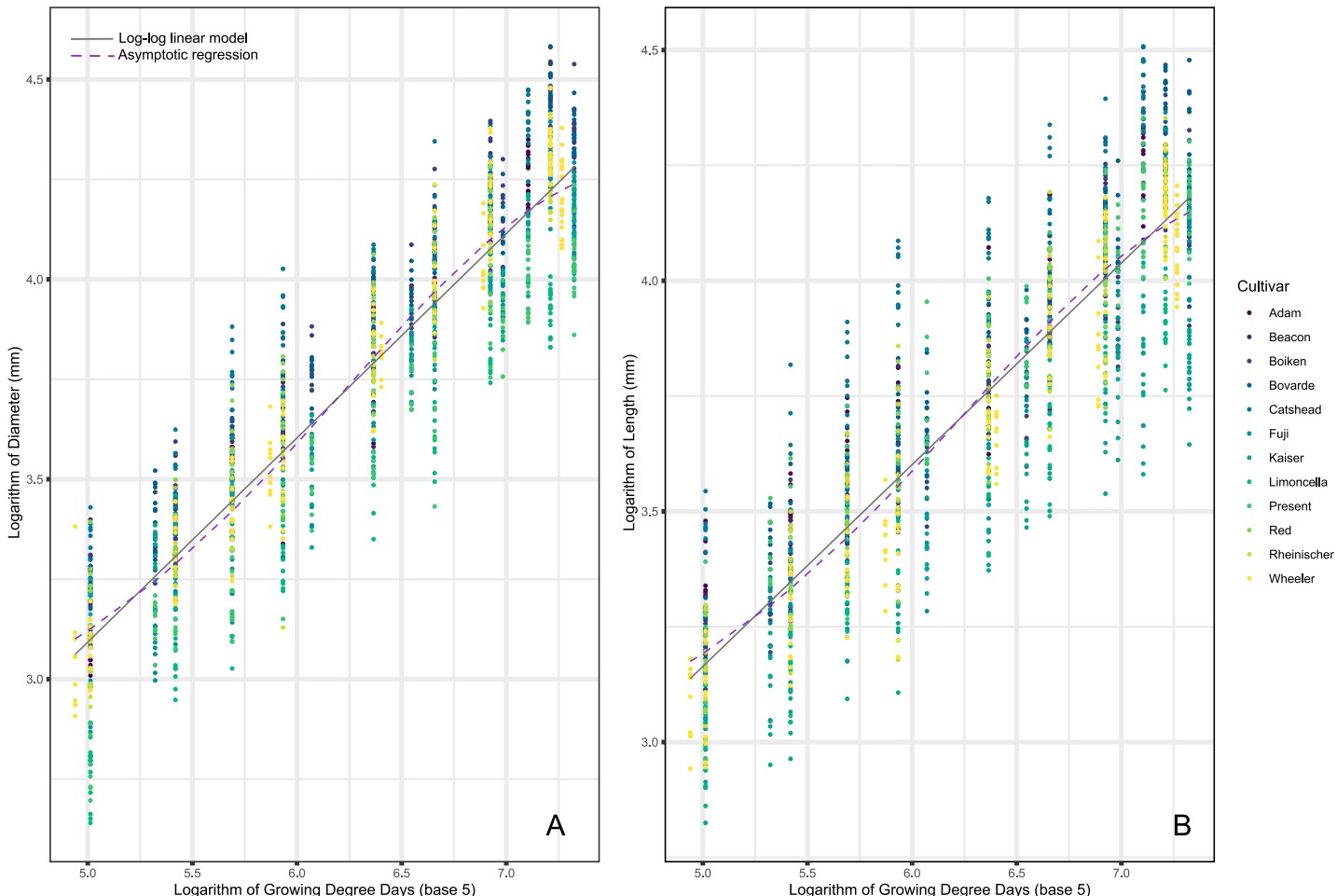

**Fig 1. Model fit for log-log linear regression (solid grey) and asymptotic regression (dashed purple) between growing degree days and measured morphometrics (A: Diameter, B: Length).** Twelve apple cultivars were sampled throughout the 2013 growing season, with six of them repeated in the 2014 growing season. Both growing degree days and respective morphometrics are logarithmically transformed for illustration purposes. Growing degree days are base 5.

the strong heteroskedasticity presented in the original data, the model would not have been appropriate.

Intercept and slope estimates, as well as adjusted $R^2$ values, are presented in Table 5.

Putting these slope coefficients into context, a 10% increase in Growing Degree Days results in 1.04% increase for Length, 1.05% increase for Diameter and Centroid Size, and 1.14% increase for Weight averaged across all cultivars and both years.

## Discussion

Christodoulou et al. [1] demonstrated that for mature fruit (i.e. harvested when ripe and ready for consumption) both linear and geometric morphometrics can be used to identify apple cultivars with a 78% accuracy for a test set. In this study we explore whether these morphometrics stabilize earlier in the season. This is to ascertain whether fruit that has been picked prior to maturation can be included to train and test morphometric classification methods [26]. Here we focused on twelve cultivars, 6 of which we repeated for a second year, and studied size changes using four relevant morphometrics. Our findings demonstrate a consistent positive

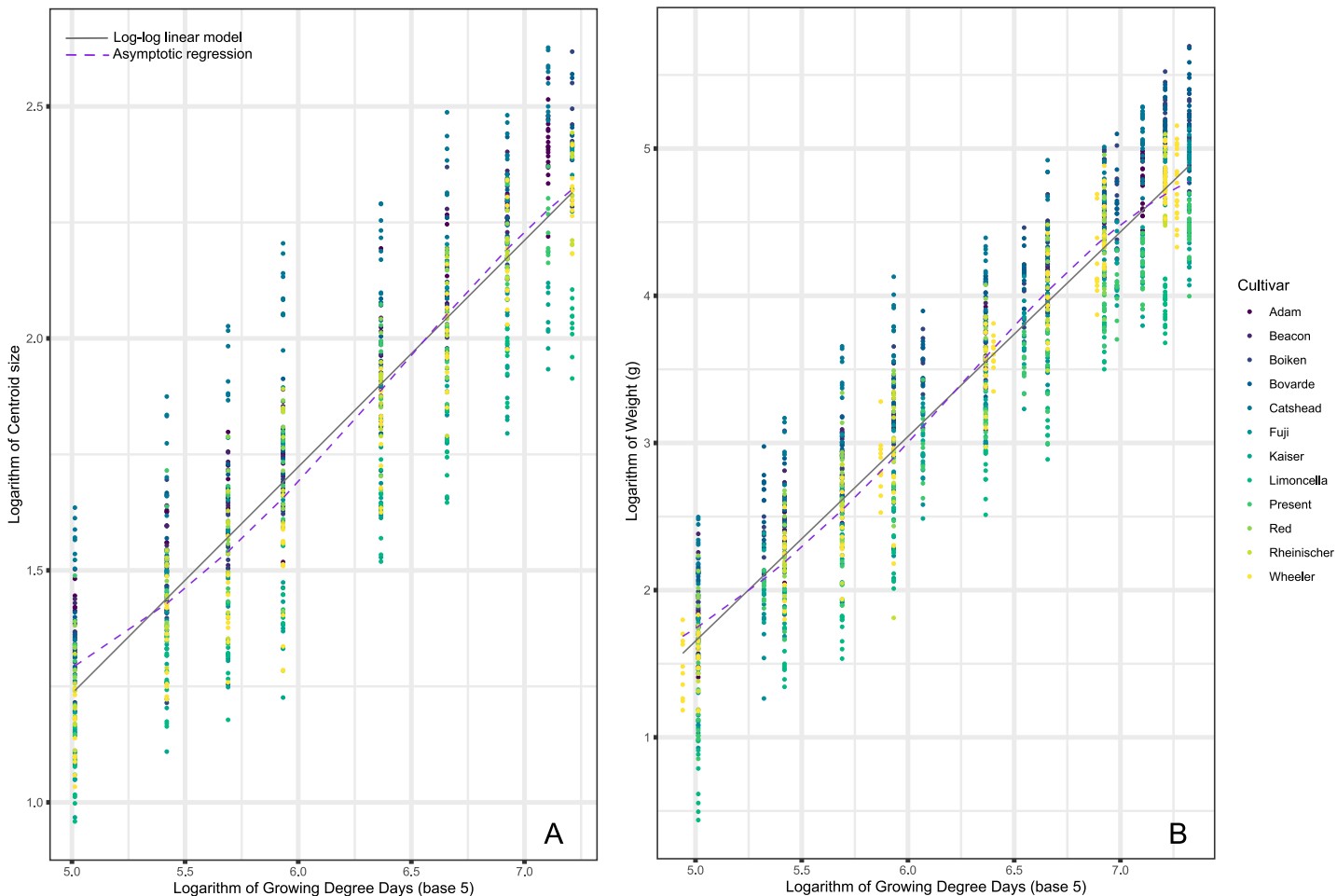

**Fig 2. Model fit for log-log linear regression (solid grey) and asymptotic regression (dashed purple) between growing degree days and measured morphometrics (A: Centroid Size, B: Weight).** Twelve apple cultivars were sampled throughout the 2013 growing season, with six of them repeated in the 2014 growing season. Both growing degree days and respective morphometrics are logarithmically transformed for illustration purposes. Growing degree days are base 5.

correlation between size and growing degree days—our proxy for timing in the growing season. This suggests that the difference in size metrics between fruit sampled earlier and matured in storage and those ripened on the tree would depend heavily on how much earlier the fruit were picked.

For cultivars that store well, harvesting practices need to be adapted to enhance the storage potential of the fruit [27]. The main adaptation in harvesting, if an apple is to be stored, is timing [28]. The general practice in the industry is that the longer the fruit is to be stored, the

**Table 4. Bayesian Information Criterion (BIC) for each fit.**

|  | Length | Diameter | Weight | Centroid size |
|---|---|---|---|---|
| Log-log linear | -937.30 | -1038.66 | 1496.14 | -626.46 |
| Asymptotic | -924.38 | -1021.05 | 1531.25 | -619.84 |

For all four morphometrics, there is strong evidence that the log-log linear regression is a more appropriate model choice when compared to an asymptotic regression.

Table 5. Summary results from log-log linear regression models.

|  | Intercept | Slope | Adjusted R$^2$ |
|---|---|---|---|
| Length | 0.978 | 0.437 | 0.783 |
| Diameter | 0.537 | 0.511 | 0.841 |
| Weight | -5.294 | 1.389 | 0.867 |
| Centroid Size | -1.208 | 0.488 | 0.803 |

Intercept, Slope, and adjusted R$^2$ for log-log linear models between Length, Diameter, Weight, or Centroid Size, and Growing Degree Days (base 5).

earlier it needs to be harvested. Timing differences between early harvest and ripened-on-tree harvest can range from three days to approximately two weeks [29].

For the two years under study, two weeks prior to the last harvest date is equivalent to a 10% reduction of growing degree days. For all the fitted models this is estimated to approximate to a 1% reduction in morphometrics. This suggests that within the context of apple storage, earlier harvest does not lead to substantially smaller fruit.

Our findings also resolve the discrepancy in literature between linear growth patterns and sigmoid growth. The plateau of the sigmoid fit, as suggested by multiple authors [9,11–13] is also present in our model, although less evident as we are using a two logarithmic transformations. In terms of the slow growth stage for the very beginning of the season, we did not observe an obvious early plateau such as the one described by Denne [9]. We believe this to be due to our collection timings and the fact that our sampling strategy substantially differed from the one described by Denne. Denne hand-pollinated her samples and could therefore follow their development even before petal-drop. We relied on natural pollination, meaning we only sampled after fruitlets formed. That is a substantial difference that explains the absence of early plateau in our findings. We believe our log-log model to be in line with the sigmoid models presented by the authors mentioned above. Using two logarithmic transformations, a simple linear model fits the data.

We substantially diverge however from the linear and expo-linear models on untransformed data proposed by Tukey and Young [14], Blanpied [15], Lakso et al. [16], and Saei et al. [18]. In the case of Tukey and Young, their suggestion of a linear model does not align with their own graphical representations where a plateau is clearly visible. Blanpied [15] described observations from average measurements without giving any indication as to the variation at each time point. For Lakso et al. [16], the proposal of fitting an exponential model in the early stages followed by a linear one in the later stages appears misguided as during the later stages there is an evident non-normality of errors indicated by their graphics. This makes the linear model for the second stage a poor fit. Finally, Saei et al [18] employing an adapted model from Buchwald and Sveiczer [19,20], fail to provide sufficient information to allow the reader to evaluate the appropriateness of their recommended approach.

We conclude that our log-log linear regression matches some of the published literature recommending asymptotic or sigmoid growth. Through the use of a double log transformation we present a more simplified modeling approach, with fewer parameters. For the purposes of exploring the impact of early harvest on size morphometrics, we demonstrate that a two-week earlier harvest time, such as the one regularly used in the supermarket supply chain, would be equivalent to approximately 1% reduction in size.

## Conclusions

In this work we explored the developmental process of apples during the growing season with respect to size morphometrics, specifically weight, length, diameter, and centroid size. We consistently found them to be adequately modelled against growing degree days using a linear regression after logarithmic transformations for both predictive and explanatory variables. The fitted models suggest a positive correlation between the variables, where a 10% increase in growing degree days is associated with a 1% increase for the predicted morphometric. As such we believe our findings to be in line with previous work presenting a plateau in later stages of development and that a two-week earlier sampling for storage purposes does not substantially alter the size of the fruit. This is important for both fruit sellers and buyers who may be paying a premium for particular cultivars. Apple cultivars whether harvested fully ripe or up to two weeks prior to maturity, demonstrate the characteristic features that distinguish them.

## Acknowledgments

We thank Prof Nick Battey, Dr Matthew Ordidge, and the National Fruit Collection, Brogdale, Kent, for access to samples and assistance with sampling.

## Author Contributions

**Conceptualization:** Alastair Culham.

**Data curation:** Maria D. Christodoulou.

**Formal analysis:** Maria D. Christodoulou.

**Funding acquisition:** Alastair Culham.

**Investigation:** Maria D. Christodoulou.

**Methodology:** Maria D. Christodoulou, Alastair Culham.

**Project administration:** Alastair Culham.

**Supervision:** Alastair Culham.

**Validation:** Maria D. Christodoulou.

**Visualization:** Maria D. Christodoulou, Alastair Culham.

**Writing – original draft:** Maria D. Christodoulou, Alastair Culham.

**Writing – review & editing:** Maria D. Christodoulou, Alastair Culham.

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
