## [Decision Letter · Decision Letter 0]

1 Feb 2021

PONE-D-20-33261

When do apples stop growing, and why does it matter?

PLOS ONE

Dear Dr. Culham,

Thank you for submitting your manuscript to PLOS ONE. After careful consideration, we feel that it has merit but does not fully meet PLOS ONE’s publication criteria as it currently stands. Therefore, we invite you to submit a revised version of the manuscript that addresses the points raised during the review process.

We look forward to receiving your revised manuscript.

Kind regards,

Zhenhai Han, PhD

Academic Editor

PLOS ONE

Journal Requirements:

Reviewers' comments:

Reviewer's Responses to Questions

**Comments to the Author**

1. Is the manuscript technically sound, and do the data support the conclusions?

Reviewer #1: Yes

Reviewer #2: Yes

2. Has the statistical analysis been performed appropriately and rigorously? 

Reviewer #1: Yes

Reviewer #2: Yes

3. Have the authors made all data underlying the findings in their manuscript fully available?

Reviewer #1: Yes

Reviewer #2: Yes

4. Is the manuscript presented in an intelligible fashion and written in standard English?

Reviewer #1: Yes

Reviewer #2: Yes

5. Review Comments to the Author

Reviewer #1: I have reviewed the submission "When do apples stop growing, and why does it matter?" authored by Maria D Christodoulou and colleague. In this manuscript, the authors explored the developmental process of apples during the growing season with respect to size morphometric, specifically weight, length, diameter, and centroid size and a new ‘Log-log linear’ model is proposed according to these data. Importantly, a positive correlation between the variables, where a 10% increase in growing degree days is associated with a 1% increase for the predicted morphometric, can be found by this model. It will be available to commercial distribution of apples.

The manuscript is concise and well written, but I have a few questions about it.

1. As we known, biennial bearing often occurs in fruit trees and also affects fruit size. How about the fruit yield of the experimental trees at different years?

2. I suggest that authors should add some detailed information about the meaning of different growth curve (Linear, Sigmoid, Curvilinear or else mentioned in this research) in the introduction.

3. If author can give a discussion about the limitation of this model.

Reviewer #2: The manuscript which is named “When do apples stop growing, and why does it matter?” analyzed twelve apple cultivars at regular time points throughout the growing season with four morphometrics, and finally get a conclusion that a 10% increase in growing degree days is associated with a 1% increase for the predicted morphometric. This research is very interesting which could provide suggestions when to pick apples to balance the storage and maturity. Thus, I recommend this work should be accepted for publication, but the results parts wrote too simplified and should be improved.

6. PLOS authors have the option to publish the peer review history of their article (what does this mean?). If published, this will include your full peer review and any attached files.

Reviewer #1: **Yes: **Tong Li

Reviewer #2: No

---

## [Author Response · Author response to Decision Letter 0]

12 Mar 2021

Response to Reviewers:

Reviewer #1: I have reviewed the submission "When do apples stop growing, and why does it matter?" authored by Maria D Christodoulou and colleague. In this manuscript, the authors explored the developmental process of apples during the growing season with respect to size morphometric, specifically weight, length, diameter, and centroid size and a new ‘Log-log linear’ model is proposed according to these data. Importantly, a positive correlation between the variables, where a 10% increase in growing degree days is associated with a 1% increase for the predicted morphometric, can be found by this model. It will be available to commercial distribution of apples.

The manuscript is concise and well written, but I have a few questions about it.

1. As we known, biennial bearing often occurs in fruit trees and also affects fruit size. How about the fruit yield of the experimental trees at different years?

We thank reviewer #1 for their questions. Biennial patterns were a primary motivation for running the trial for two consecutive years for half the cultivars in our study. As part of our preliminary analysis, we did compare the weight of the fruit in the harvest weeks for each cultivar between the two years using a Welch’s 2 sample t-test and found no significant differences between the two years. We have now updated our code script to include this step of our preliminary analysis. We have also amended lines 185-188 of our manuscript to state this. What we did not record during the sampling was whether there were overall substantial differences in fruit numbers on the trees between the years and therefore we cannot comment on the overall yield. 

2. I suggest that authors should add some detailed information about the meaning of different growth curve (Linear, Sigmoid, Curvilinear or else mentioned in this research) in the introduction.

We have included a paragraph with a more detailed description of the overall shapes presented in the literature in lines 113-124 of our Introduction.

3. If author can give a discussion about the limitation of this model.

We have included a segment on the weak points of the log-log linear regression in lines 223-228 as per the reviewer’s suggestions.

Reviewer #2: The manuscript which is named “When do apples stop growing, and why does it matter?” analyzed twelve apple cultivars at regular time points throughout the growing season with four morphometrics, and finally get a conclusion that a 10% increase in growing degree days is associated with a 1% increase for the predicted morphometric. This research is very interesting which could provide suggestions when to pick apples to balance the storage and maturity. Thus, I recommend this work should be accepted for publication, but the results parts wrote too simplified and should be improved.

We thank reviewer #2 for their feedback and recommendation for publication. We are not clear as to what their precise feedback on the improvement of our results is as they follow closely the steps of our analysis.

---

## [Decision Letter · Decision Letter 1]

14 May 2021

When do apples stop growing, and why does it matter?

PONE-D-20-33261R1

Dear Dr. Culhan,

We’re pleased to inform you that your manuscript has been judged scientifically suitable for publication and will be formally accepted for publication once it meets all outstanding technical requirements.

Kind regards,

Molly M. Jahn

Academic Editor

PLOS ONE

Additional Editor Comments (optional):

Reviewers' comments:

Reviewer's Responses to Questions

**Comments to the Author**

1. If the authors have adequately addressed your comments raised in a previous round of review and you feel that this manuscript is now acceptable for publication, you may indicate that here to bypass the “Comments to the Author” section, enter your conflict of interest statement in the “Confidential to Editor” section, and submit your "Accept" recommendation.

Reviewer #1: All comments have been addressed

Reviewer #2: All comments have been addressed

2. Is the manuscript technically sound, and do the data support the conclusions?

Reviewer #1: Yes

Reviewer #2: (No Response)

3. Has the statistical analysis been performed appropriately and rigorously? 

Reviewer #1: Yes

Reviewer #2: (No Response)

4. Have the authors made all data underlying the findings in their manuscript fully available?

Reviewer #1: Yes

Reviewer #2: (No Response)

5. Is the manuscript presented in an intelligible fashion and written in standard English?

Reviewer #1: Yes

Reviewer #2: (No Response)

6. Review Comments to the Author

Reviewer #1: (No Response)

Reviewer #2: (No Response)

7. PLOS authors have the option to publish the peer review history of their article (what does this mean?). If published, this will include your full peer review and any attached files.

Reviewer #1: **Yes: **Tong Li

Reviewer #2: No

---

## [Editor Report · Acceptance letter]

28 May 2021

PONE-D-20-33261R1 

When do apples stop growing, and why does it matter? 

Dear Dr. Culham:

I'm pleased to inform you that your manuscript has been deemed suitable for publication in PLOS ONE. Congratulations! Your manuscript is now with our production department. 

Kind regards, 

on behalf of

Dr. Molly M. Jahn 

%CORR_ED_EDITOR_ROLE%

PLOS ONE